# Implementing digital computing with DNA-based switching circuits

Fei Wang [1,2,7], Hui Lv[3,4,7], Qian Li[1], Jiang Li[3,5], Xueli Zhang[2], Jiye Shi [1], Lihua Wang[3,5,6]* & Chunhai Fan [1]*

DNA strand displacement reactions (SDRs) provide a set of intelligent toolboxes for developing molecular computation. Whereas SDR-based logic gate circuits have achieved a high level of complexity, the scale-up for practical achievable computational tasks remains a hurdle. Switching circuits that were originally proposed by Shannon in 1938 and nowadays widely used in telecommunication represent an alternative and efficient means to realize fast-speed and high-bandwidth communication. Here we develop SDR-based DNA switching circuits (DSCs) for implementing digital computing. Using a routing strategy on a programmable DNA switch canvas, we show that arbitrary Boolean functions can be represented by DSCs and implemented with molecular switches with high computing speed. We further demonstrate the implementation of full-adder and square-rooting functions using DSCs, which only uses down to 1/4 DNA strands as compared with a dual-rail logic expression-based design. We expect that DSCs provide a design paradigm for digital computation with biomolecules.

[1] School of Chemistry and Chemical Engineering, Institute of Molecular Medicine, Renji Hospital, School of Medicine, Shanghai Jiao Tong University, Shanghai 201240, China. [2] Joint Research Center for Precision Medicine, Shanghai Jiao Tong University and Affiliated Sixth People's Hospital South Campus, Southern Medical University Affiliated Fengxian Hospital, Shanghai 201499, China. [3] Division of Physical Biology, CAS Key Laboratory of Interfacial Physics and Technology, Shanghai Institute of Applied Physics, Chinese Academy of Sciences, Shanghai 201800, China. [4] University of Chinese Academy of Sciences, Beijing 100049, China. [5] Shanghai Synchrotron Radiation Facility, Zhangjiang Laboratory, Shanghai Advanced Research Institute, Chinese Academy of Sciences, Shanghai 201210, China. [6] Shanghai Key Laboratory of Green Chemistry and Chemical Processes, School of Chemistry and Molecular Engineering, East China Normal University, 500 Dongchuan Road, Shanghai 200241, China. [7]These authors contributed equally: Fei Wang, Hui Lv.
*email: wanglihua@sinap.ac.cn; fanchunhai@sjtu.edu.cn

DNA strand displacement reactions (SDRs)[1–3] have been employed to implement highly complex tasks such as molecular computing[4,5], information processing[6–8], and nanorobots[9–11]. In molecular computing, molecular circuits operate by the action of orthogonal molecules[12]. Hence, when the orthogonality decreases with circuit scale[13], the actual achievable size is restricted, which is different with electronic circuits in which all elements are driven by the same electric signal. SDR enables an enzyme-free implementation strategy to develop molecular circuits for arithmetical purpose[14,15], which offers high selection space of orthogonal molecules[16].

Recent advances in modular molecular design and programmable reaction dynamics have led to the development of SDR-based Boolean logic gates with unparalleled complexity[5,13,17–19]. However, limitations associated with logic gate circuits emerges, which hinders the scale-up for practical achievable computational tasks as follows. (i) Typical approaches to increase the scalability involves the combination of basic gates such as AND and OR[5,13,20,21], which nevertheless increases the use of gates[22–26]. (ii) Using uniform motif and implementing AND and OR gate with different threshold concentrations have shown high modularity and potential for constructing complex computing networks. However, the thresholding and amplifying process generally poses a limit on the computing speed[5]. (iii) NOT logic is difficult to implement experimentally when the ON or OFF state of an input is assigned with the presence or absence of the molecule[13]. As the NOT gate started produce ON signals before receiving the output of its upstream gate, the circuit might generate a false output.

During the early stage of electronic circuits, Shannon[27] established a systematic symbolic approach to design relay contacts or switching circuits (SCs). In theory, any digital circuit can be represented by a set of equations, the terms of the equations corresponding to certain switches in the circuits. SCs can implement arbitrary digital functions in an efficient and economical manner. For example, SCs form the basis of modern telecommunication that is characteristic of fast speed and high bandwidth. Previous work has theoretically proposed the implementation of SCs with "seesaw" gates[28]. Here we develop an SDR-based strategy to experimentally realize DNA-based SCs (DSCs) for molecular digital computing. In a typical DSC, switches act as the sole basic functional element in a circuit, the molecular design is thus uniform, which allows high modularity, programmability, and scalability. The direction of current signal transmission can be finely controlled by the difference in the free energy of molecules in the reaction pathway. We also expect the increase of the computing speed with the reduced use of orthogonal molecules.

## Results
**DSC implementation scheme**. A typical electronic SC consists of switches that connects or disconnects two nodes depending on the input voltage[29]. Here we designed molecular switches based on SDR, to emulate such functions for constructing DSCs (Fig. 1a). In this design, a DSC contains up to three types of molecular elements. A starting switch at the entrance of the circuit responds to its switching signal and generates a current signal. A downstream switch responds to the current signal and switching signal, producing a current signal. Finally, a reporter responds to the current signal and generates fluorescent output. The response to a switching signal is realized with a molecular switch as shown in Fig. 1b. Molecular implementation of the switch function is realized by toehold-mediated SDR. A single-stranded DNA acting as a switching signal binds to the double-stranded switch molecule via a 5 nt toehold[26], displaces

the current signal strand, and frees it in an entropy-driven manner. The released current signal is transmitted to a downstream switch to activate the responsive function of the downstream switch (Fig. 1c). We designed the downstream switch molecule with two binding domains: C domain to respond to a current signal and S domain to respond to a switching signal. The switch cannot flip to the ON state in the absence of the current signal. The transmission of the current signal from an upstream switch to a downstream switch is achieved by the hybridization of the current signal with the downstream switch through C domain, exposing the toehold for the binding of switching signal.

Boolean functions were realized with DSCs by programming the connection topologies of molecular switches. Switching signals acted as inputs for Boolean computation. The AND logic was implemented with two switches in series (Fig. 1d). Two switches in parallel performed OR logic (Fig. 1e). The complex function OR(NOR(a,b),c) containing a NOR gate and an OR gate (Fig. 1f, left) was compiled to dual-rail representation (Fig. 1f, middle) with AND and OR gates. This function was realized with a DSC with only three switches (Fig. 1f, right).

**Switch flipping and current signal transmission**. We first experimentally tested the switching and signal transmission functions using a one-switch circuit and a circuit with two series switches, respectively. The signal transmission process in the series circuit is shown in Fig. 2a. The circuit contained an upstream switch x and a downstream switch y. Initially, switch x was OFF and switch y was blocked as the current signal had not arrived. Switching signal x flipped switch x to the ON state and released a current signal. The current signal was then transmitted to switch y, allowing switch y to respond to its switching signal. The current signal generated by switch y was read out by a reporter. Corresponding SDRs are illustrated in Fig. 2b. Single-stranded x representing a switching signal bound to switch S(x), displaced the current signal strand and released it. As a result, switch x jumped from OFF to ON state (fully double-stranded), generating a current signal. The molecule acting as a downstream switch y was named CS(y) with a C domain in addition to the S domain. Current signal from switch x interacted with CS(y) and switch y was activated by exposing the middle toehold. Then, switching signal y bound to switch y via the exposed toehold, generating a current signal. Finally, the reporter molecule converted molecular concentration of the current molecule from switch y to fluorescence intensity using a fluorophore-quencher pair.

The one-switch circuit was tested with and without the addition of switching signal and it gave the correct outputs. The switch was at 1× concentration, where 1× was 100 nM (Supplementary Fig. 1) and switching signal was at 2× concentration to achieve fast and complete flipping (Supplementary Fig. 3). In the experiments, all the circuit components, except the switching signal strand, were mixed on ice according to the optimized molar ratio of components in TE buffer with 12.5 mM $MgCl_2$. Reaction temperature was kept at 20 °C (±2° C) (Supplementary Fig. 7). We observed a quick increase of output signal with the addition of switching signal within 16 min ($t_{1/2} = 1$ min) (Fig. 2c). In contrast, the output remained low without input (<5%). These results suggested that the DNA switch responded effectively to its switching signal.

For the circuit with two series switches, upstream switch x was at 1.5× and downstream switch y was at 1×, where 1× was 100 nM. These two concentrations were used to ensure that the upstream switch could produce sufficient output for the downstream switch (Supplementary Fig. 2). To ensure that the every switch could be opened, we added sufficient input strands (2×).

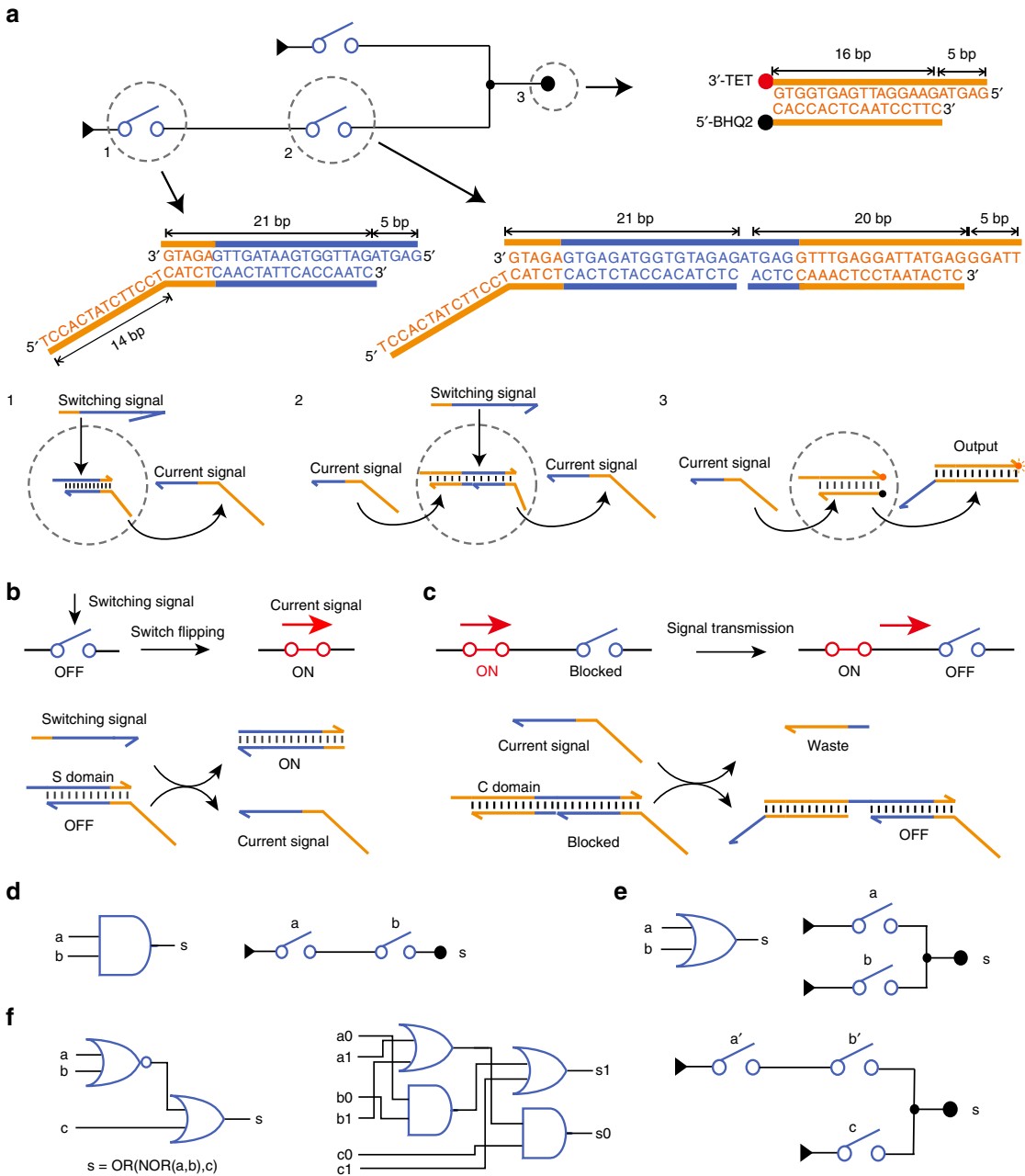

**Fig. 1 DSC-based implementation of Boolean computation. a** Schematic illustration and molecular details of the DSC system. Typically, a DSC contains three types of elementary molecules: (1) Starting switch that responds to its switching signal. (2) Downstream switch that responds to current signal and switching signal. (3) Reporter that responds to current signal and generates fluorescent output. **b** Schematic illustration and DNA implementation for switching signal response. **c** Schematic illustration and DNA implementation for current signal transmission. **d** AND logic and the corresponding SC. **e** OR logic and the corresponding SC. **f** Left: a NOR-OR circuit with NOT operation. Middle: dual-rail representation of the NOR-OR circuit. Right: SC implementation, where a' and b' represent the complementary value of a and b, respectively.

The essential strategy was that the concentration of a switch was determined such that this switch could always generate sufficient output strands for its downstream switches. This strategy for determination of concentrations in cascade was applied to all circuits. The measurements of the two-switch circuit showed high output fluorescence at the presence of two switching signals ($t_{1/2} < 3$ min) (Fig. 2d). When only switching signal x or y was added, we observed low output fluorescence (<5%). These results are consistent with the designed function of series switches, with a computing speed much faster than reactions using logic gates in solution (several hours)[1,4,13].

**Fan-out and fan-in of DNA switches**. With the programmability of C domain on downstream switch molecules, the current signal from an upstream switch can be easily transmitted to multiple downstream switches (fan-out). We designed and tested a DSC with a two-output switch (Fig. 3a). As shown in Fig. 3b, the same DNA sequence of C domain (shown in red) on CS(x) and CS(y) allowed current transmission path of w–x and w–y. Current signal from S(w) interacted with both C domains that blocked switch x and y, resulting in activation of the two switches. To overcome signal decay, concentration ratio of upstream and downstream was optimized (Supplementary Fig. 4). We found

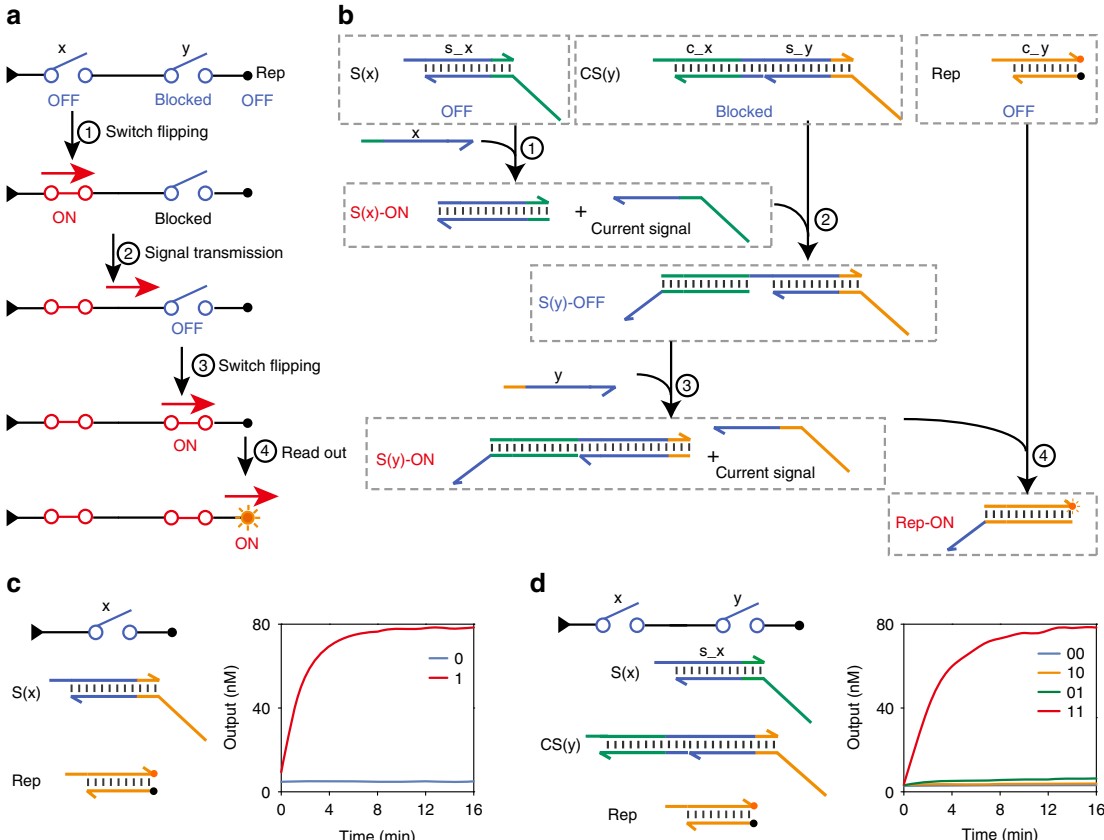

**Fig. 2 Experimental implementation of switch flipping and current signal transmission. a** Schematic illustration of signal propagation in a switching circuit with two series switches. **b** Molecular implementation and chemical reaction network of the circuit in **a**. **c** Molecular implementation and fluorescence kinetics data of a single-switch circuit. **d** Molecular implementation and fluorescence kinetics data of a two-switch circuit. Source data are provided as a Source Data file.

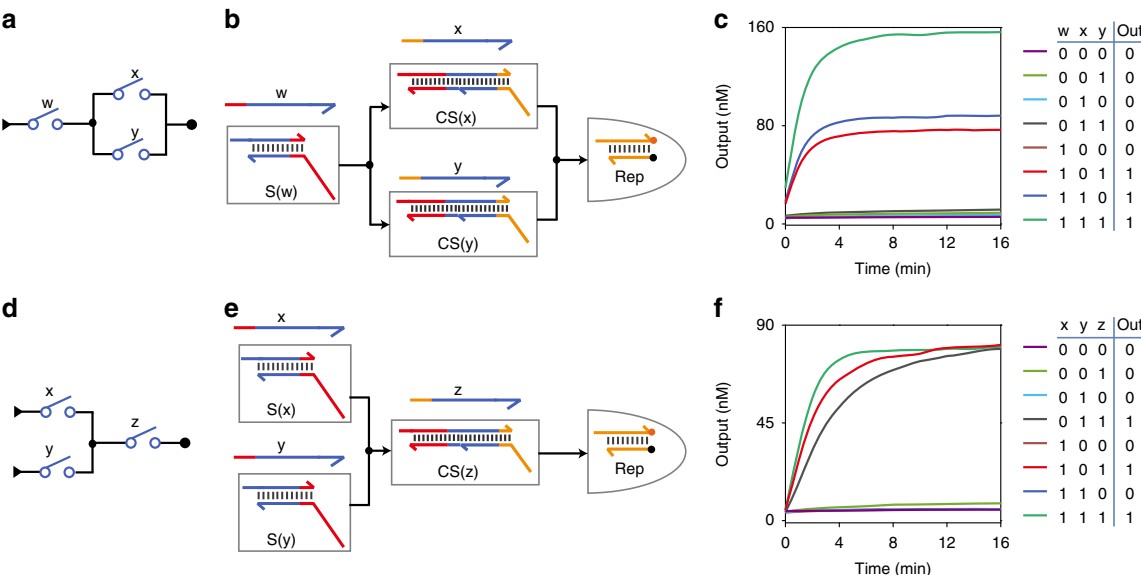

**Fig. 3 Fan-out and fan-in of DNA switches. a–c** Switching circuit diagram (**a**), molecular implementation (**b**), and fluorescence readout (**c**) of a circuit with a two-output switch. **d–f** Switching circuit diagram (**d**), molecular implementation (**e**), and fluorescence readout (**f**) of a circuit with a two-input switch. Source data are provided as a Source Data file.

that a ratio of 3:1 satisfied sufficient output of both downstream switches ($t_{1/2} < 3$ min; Fig. 3c).

With the programmability of current signal strand on upstream switches, DSCs here can easily support fan-in that represents current signals from multiple upstream switches enter one downstream switch (Fig. 3d). As shown in Fig. 3e, current signals from upstream switches x and y had the same sequence and each of them was able to interact with C domain on CS(z); thus, current signals from both S(x) and S(y) were transmitted to switch z. Fluorescence kinetics showed that either input combination x–z or y–z lead to high output ($t_{1/2} < 3$ min, Fig. 3f). This circuit also had the same function of logic gate circuit AND (OR(x,y),z), whereas the computing speed was much faster than the reported two-layer logic gate circuits in solution[22,26]

With the abilities to perform basic functions including cascading, fan-in, and fan-out, theoretically the molecular switches can be used to build DSCs with arbitrary topologies. We next explored the circuit design strategy to perform arbitrary Boolean functions. Inspired by the circuit proposed by Shannon[27] to solve symmetric functions, we developed a programmable switch canvas with complementary switch pairs (Fig. 4a, middle). On the canvas, the connection between adjacent switches is configurable. A Boolean function is represented by a DSC by programmable routing on the switch canvas. Each horizontal switch responds to a switching signal representing a positive input value (input = 1); each vertical switch responds to a switching signal representing a negative input value (input = 0). Each line of the truth table is represented as a path on the switch canvas.

As an example, we implemented a three-input voting logic by routing the switch canvas. As shown in Fig. 4a, input combination in red box in truth table was mapped on the switch canvas as the path in red. Thus, the layer number of the corresponding DSC was no more than the number of variables. Then the generated DSC was simplified (Supplementary Fig. 9) and the switches were numbered for further molecular implementation. We designed the sequence for each molecule according to requirements for cascade, fan-out, and fan-in as discussed above. Using this strategy, the three-input voting function was demonstrated with a three-layer six-switch DSC (Fig. 4b). For comparison, this function was also realized using an AND–OR logic gate circuit with six gates, with the same molecular design as the DSC (Fig. 4e). We investigated the change of free energy during the reaction process for the DSC and logic gate circuit. We found that the magnitude of stepwise free-energy decrease is very close for all input combinations that lead to output in the DSC (Fig. 4c). In contrast, the logic gate circuit presented different trends of energy change for the four input combinations (Fig. 4f). Previous study has shown that the reaction rate of SDR is positively correlated to energy change[30,31]. We hypothesize that the required time for current signal to arrive to reporter was less diverged using the DSC. As a result, the computing speed was improved. Numerical simulations (Supplementary Fig. 10) confirmed that the DSC performed faster than the logic gate circuit. Experimental results (Fig. 4d, g) showed a much faster computing speed for the DSC ($t_{1/2} \sim 3$ min) than the logic gate circuit ($t_{1/2} \sim 10$ min) and a lower signal leakage for outputs supposed to be 0. These results suggested that even with

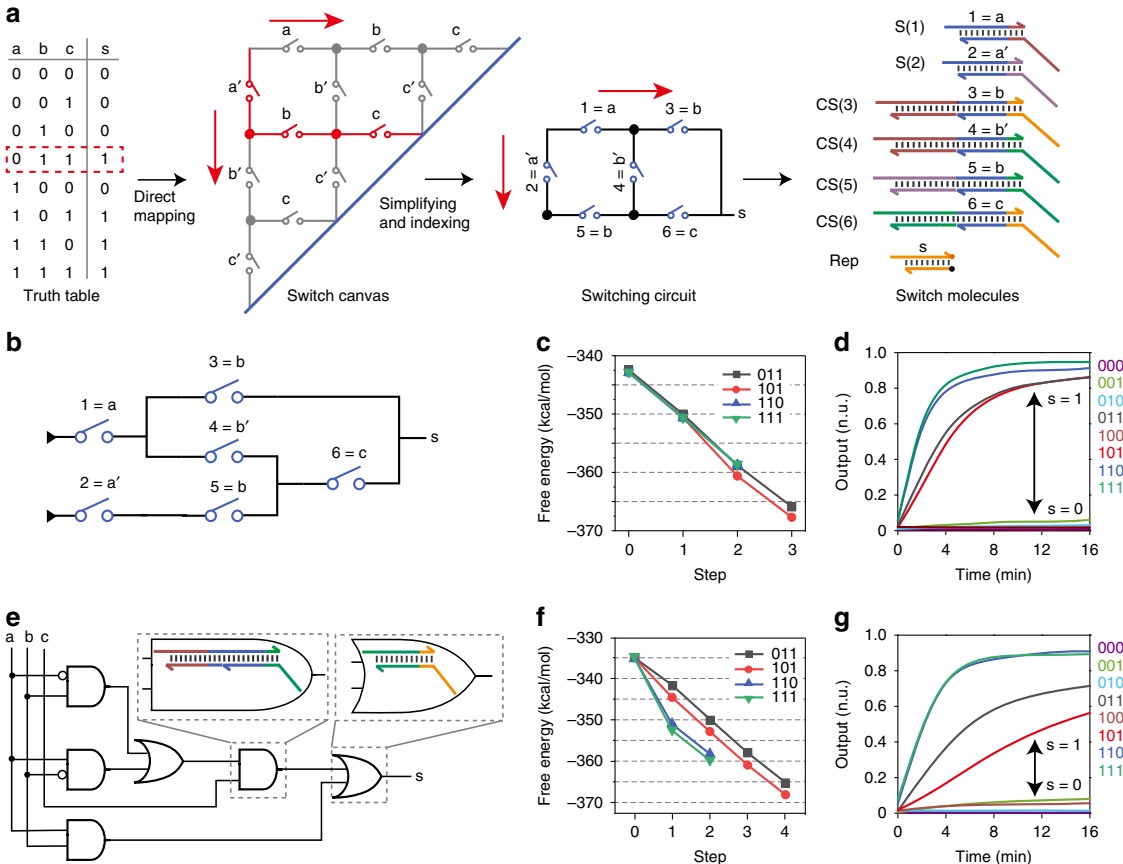

**Fig. 4 Implementing arbitrary Boolean functions with DSCs. a** The mapping process from a logic function to a DSC for a three-input voting logic. **b** DSC for the voting logic in **a**. **c** Change in free energy of stepwise switch flipping with each input combination that leads to output. **d** Experimental fluorescence readout with DSC. **e** Logic circuit implementation of the voting logic in **a**. **f** Change in free energy of stepwise gate opening with each input combination that leads to output. **g** Experimental fluorescence readout with logic gate implementation. Source data are provided as a Source Data file.

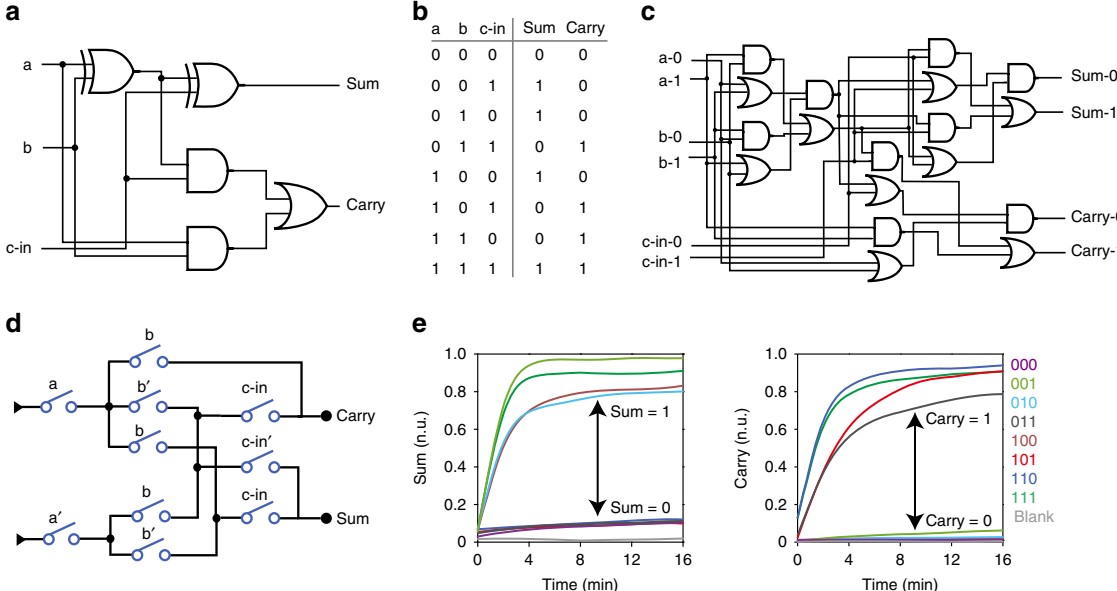

**Fig. 5 Implementing a full-adder with a DSC. a–d** Logic gate diagram (**a**), truth table (**b**), dual-rail representation (**c**), and switching circuit diagram (**d**) of the full-adder circuit. **e** Fluorescence readout of sum (left) and carry (right) with all possible combinations of inputs. Source data are provided as a Source Data file.

the same molecular design, DSC showed higher computing speed and signal-to-noise ratio (SNR) than logic gates for some functions.

**Implementing a full-adder with a DSC**. To exemplify arbitrary digital computing using DSCs, we further demonstrated a full-adder function. As shown in Fig. 5a, a full-adder circuit consisted two XOR logic gates, which was realized with a combination of AND, OR, and NOT gates for SDR design (Supplementary Fig. 11a). The required truth table of this function is shown in Fig. 5b. Due to the existence of NOT gate, experimental implementation of this circuit required dual-rail representation with a logic circuit consisting 18 AND or OR gates (Fig. 5c)[32,33]. The direct mapping of the truth table on the programmable switch canvas generated a three-layer ten-switch DSC to implement the full-adder function (Fig. 5d).

The digital computing was performed with ten switch molecules and two reporter molecules to read out Carry and Sum output. With all possible input combinations, outputs of Sum and Carry went to the correct ON or OFF state ($t_{1/2} < 3$ min) (Fig. 5e). When it runs, there are up to 42 different DNA strands interacting in one test tube. Without the demand for dual-rail representation, the number of involved DNA molecules was decreased by nearly half than that of logic gate implementation (Supplementary Fig. 11b).

**A DSC for fast computation of square root**. To further demonstrate the power of DSCs for performing complex digital tasks, we designed a DSC to compute the floor of the square root of a four-bit input (Fig. 6a). Compared with a dual-rail logic gate circuit to perform the same function (Fig. 6b), the DSC showed a much simpler layout with only two layers. We then compared our DSC with the previous ones using seesaw gates and single-stranded gates to solve four-bit square-rooting. With seven switches, our DSC showed a reduction in the number of computing elements (Fig. 6c). Although the seesaw gate circuit based on toehold-mediated SDR had around 100 strands in the reaction[13], the DSC had at most 24 strands participating one computation, showing a nearly 75% reduction in required strands

(Fig. 6d). Particularly, in comparison with the recently proposed single-stranded gates using strand-displacing polymerase that dramatically reduced strand complexity (with ~37 strands)[34], our DSC used even less strands, even with double-stranded switches, suggesting the simplicity in molecular reaction with the SC architecture.

We tested all 16 possible four-bit inputs and all the computations gave correct outputs (Supplementary Fig. 12). The computing kinetics of four representative inputs that led to different outputs were shown in Fig. 6e. The circuit with each input computed the outputs in a half-completion time of <10 min, which showed, to our knowledge, the fastest computing with diffusive components. Analyzing the trends of leakage and output with number of participated switches based on the experimental results, we estimate more than 30 switches were allowed to participate in a single reaction at the threshold of 40% (Supplementary Fig. 13), making it possible to feasibly implement more complex digital algorithms with DSCs.

## Discussion

Here we report a general strategy for modular design of DSCs to implement arbitrary digital DNA computing. The modularity can be reflected in molecular structure design, sequence design, and DSC layout generation. The rule for molecular structure design is simple: an upstream switch only has an S domain, whereas a downstream switch has a C domain and an S domain. Sequence design constraints of C domain are determined by the adjacent connection patterns (Supplementary Fig. 8). The DSC layout is generated by modularly mapping input combinations in the truth table to a transmission pathway on switch canvas. Therefore, any desired functions of given variables can be realized with corresponding DSCs by programming the connection of switches. Particularly, when there is NOT logic in a circuit, DSC provides a more efficient implementation. Although the dual-rail expression provides a solution to the experimental realization of NOT-containing logic circuits, the size of the circuit needs to be doubled. Unlike electronic circuits in which all elements are driven by the same electric signal, molecular circuits operate by the action of orthogonal molecules. As the orthogonality decreases with the

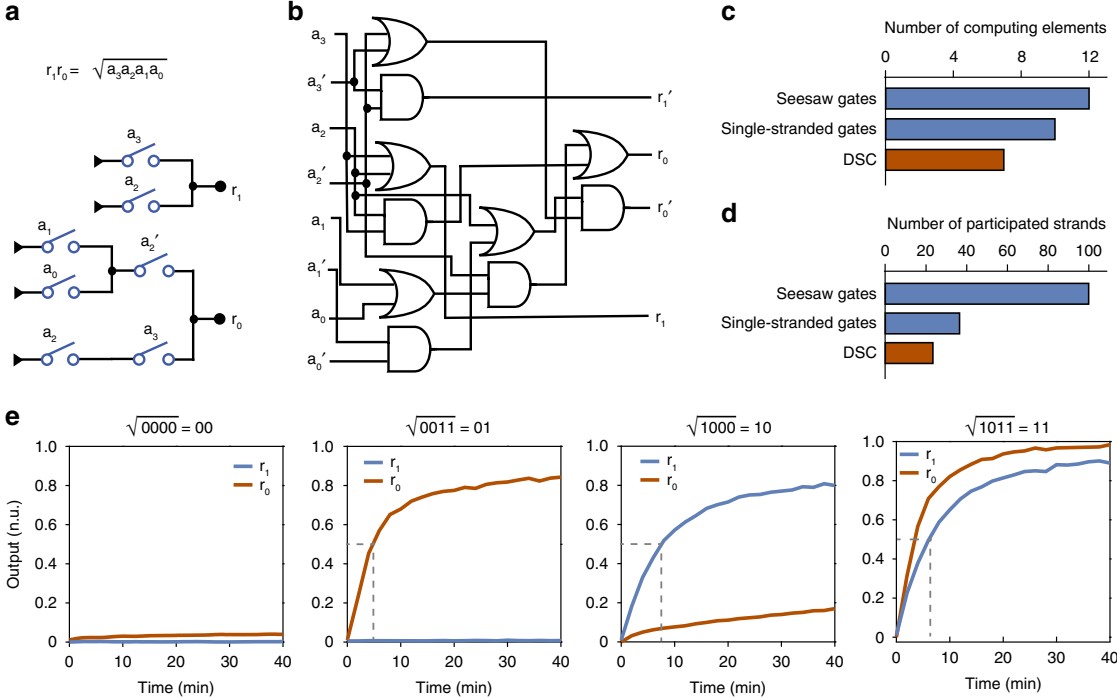

**Fig. 6 A DSC for square-root calculation. a** The DSC used to calculate the square root of a four-bit number. **b** A dual-rail logic circuit to perform the same square-rooting function. **c** Number of computing elements using DSC in comparison with previous logic gate circuits with seesaw gates[13] and single-stranded gates[34]. **d** Number of participated DNA strands using DSC in comparison with previous logic gates implementations. **e** Experimental computing kinetics with four representative inputs. Source data are provided as a Source Data file.

circuit scale, the achievable size is restricted. Using DSCs, the required computing elements was reduced by nearly a half as compared with a dual-rail logic expression-based design, allowing for the implementation of more sophisticated functions.

DSCs can be performed much faster than previously reported molecular digital computing systems (Supplementary Table 1)[1,4,13]. This probably comes from several facts. First, as modular computing element, the switch here flips by toehold-mediated strand displacement in an entropy-driven manner, which operates fast with proper toehold and double-stranded lengths[35]. Second, by adjusting strand ratios inside a switch, leakage is effectively suppressed. This allows digital computing with high SNRs without the demand of thresholding and signal restoration process during the transmission between two layers, which helps to increase the computing speed. Third, through the mapping strategy, the depth of a generated DSC on the switch canvas is always no more than the number of inputs. Short reaction pathways allow fast reaction for all input combinations that lead to high output. In addition, a reduced number of participated DNA strands could reduce unwanted binding, making molecules enter the desired reaction pathway, which may contribute to the high speed.

In conclusion, we designed and experimentally tested SDR-based DSCs that can implement arbitrary digital functions. As originally proposed by Shannon[27], SCs support fast-speed and high-bandwidth communication, which are manifested by the reduced demand for orthogonal sequences and the increased SNR of DSCs. The improved computing capability of DSC-based systems holds great potential for molecular computers and other synthetic decision-making systems or developing nanomachines with collective behaviors.

## Methods

**Molecule design**. We used a modular strategy for generating sequences used in this study. Three types of molecular structures as shown in Fig. 1 were used:

starting switch molecules were formed by hybridization of two single strands; downstream switch molecules were formed by hybridization of three single strands; reporters were formed by single strands with fluorophore and quench pairs. Starting switch is a double strand formed by DNA hybridization of two single-stranded s1 and s2, where s1 has 26 nucleotides and s2 has 35 nucleotides. Downstream switch is also a double-stranded structure, but was formed by DNA hybridization of three single strands, in which s1 has 47 nucleotides, s2 has 35 nucleotides, and s3 has 20 nucleotides (Supplementary Fig. 6). A 5 nt toehold was chosen to initiate SDR considering specificity and reaction rate[13]. In addition, the double-helix regions were chosen to be 20 or 21 bp. To avoid unwanted secondary structures of single strands, we used three bases for sequence design to prevent the co-occurrence of G and C on the same strand. First, a library of sequences was randomly generated and screened. Then, the candidate sequences were validated by NUPACK[36] to check the binding energy and specificity. At last, these sequences were assigned to the SCs for further experiments. The detailed information of DNA sequences is given in Supplementary Tables 2–4.

**Fluorescence kinetics experiments**. Kinetic fluorescence measurements were performed using Synergy H1 Hybrid Multi-Mode Reader (BioTek) and Corning 96-well black assay plate. The temperature was kept at 20 °C throughout the reaction. Excitation and emission wavelengths used in the experiments were 510 nm and 540 nm, respectively, for TET. In general, all the circuit components, except the Input strand(s), were mixed according to the optimized molar ratio of components in TE buffer with 12.5 mM MgCl$_2$. Then, the mixture was added to the 96-well black assay plate and the initial value was recorded as baseline. The experiment was then paused for the addition of Input strand(s) and subsequent mixing by shaking. The 96-well black assay plate was then put back into the Hybrid Reader and the experiment was resumed. Fluorescence kinetics of each well was detected with intervals of 1 min or 2 min. The obtained fluorescence signal was converted to concentrations of the corresponding unquenched fluorescence strands using the calibration curve for each reporter. To construct a calibration curve, solutions containing predefined concentration of opened reporter molecules were tested (Supplementary Fig. 14). Independently repeated experiments were performed to minimize operation caused fluctuations. The fluorescence signals were linearly proportional to the concentration of opened reporter molecules (Supplementary Fig. 15) and the linear fitting result was used for signal conversion. For the readout of logic results, fluorescence intensity was normalized. The minimum level (output = 0) was determined by the minimum of all data at $t = 0$; the maximum level (output = 1) was determined by the maximum value of a parallel experiment. We used 0.4× and 0.6× as thresholds. Signals below 0.4× were treated as 0 and signals above 0.6× were treated as 1. If an output signal was between 0.4× and 0.6×, we considered it as a confusing result.

## Data availability

All the data that support the findings of this study are available within the paper and its Supplementary Information files, and from the corresponding authors upon reasonable request. Source data for Figs. 2–6 and the Supplementary Figs. are provided as a Source Data file.

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

## Acknowledgements

This work was supported by the National Key R&D Program of China (2018YFA0902600), National Natural Science Foundation of China (21834007, 21675167, 21775157, and 31571014), China Postdoctoral Science Foundation (2019M651477), the Open Large Infrastructure Research of Chinese Academy of Sciences, LU Jiaxi International Team of the Chinese Academy of Sciences, K. C. Wong Foundation at Shanghai Jiao Tong University, and the Innovative research team of high-level local universities in Shanghai.

## Author contributions

C.F. directed the research. F.W. and C.F. conceived the study. F.W., L.W., J.S., and X.Z. designed experiments. F.W., H.L., and J.L. performed the experiments and analyzed the data. F.W. performed the computer simulations. C.F., F.W., Q.L., and X.Z. wrote the manuscript. All authors discussed the results and commented on the manuscript.

## Competing interests

The authors declare no competing interests.
