## [Peer Review File · Nature Communications]

Reviewers' Comments:

Reviewer #1:

Remarks to the Author:

The manuscript "Implementing digital computing with DNA-based switching circuits" by Chunhai Fan and co-coworkers report the design of DNA switching circuits based on Strand displacement reactions. The field of DNA computing is exploding but, as the authors state in the introduction, there are still several problems to be faced. The authors try to overcome some of them by re-adapting the idea of switching circuits widely used in telecommunication. The idea is thus very interesting and quite novel. The experiments performed are very well described and convincing. Considering the topic and novelty I support the publication of this manuscript. Before that, I would suggest the authors to take care of the following minor issues:

- 1) Authors should make an effort to better clarify how their approach is improving the speed and modularity of DNA circuits. There is a nice section in the conclusion about this but my opinion is that this could be improved for clarity.
- 2) Authors should provide more experimental details in the main text. Now it seems like the concentration of the switches and inputs is not important as this is not stated during the discussion of the results. This should be more carefully described and discussed as represents an important factor in the response observed.
- 3) Authors should more clearly define the threshold levels used during their experiment
- 4) It would be also nice to provide a direct comparison with conventional DNA-based circuits in the main text rather than in the supplementary section.

Reviewer #2:

Remarks to the Author:

Wang and colleagues build several DNA-based switching circuits with multiple layers using the concept of toehold-mediated DNA strand displacement reactions, in which a single-stranded DNA molecule displaces another from a double-stranded complex by binding to an exposed short toehold domain. In the proposed switching circuits, a released strand from a starting switch is transmitted to a downstream switch, which then exposes a toehold and can therefore respond to its own switching signal strand. The idea of using DNA switching circuits for molecular digital computing (inspired by Shannon's work on electronic circuits) had been briefly proposed by Qian and Winfree (2011, *J Royal Society Interface* 8:1281-1297) as a way to build large-scale circuits. In the current manuscript, the authors designed and tested experimentally these circuits by including a reporter that responds to the signal flow and generates a fluorescent output. The study was well conducted and could demonstrate a faster computing speed and lower signal leakage in relation to logic gate circuits. However, since a similar framework to scale up and systematically create DNA circuits was suggested by other group in 2011, I would expect to see now some practical application of this research to fully meet with the high standards of *Nature Communications*.

The leakage of DNA-based switching circuits is still around 5%. Could the authors discuss what might be the sources of this leakage.

Minor points

Fig 3, panel f – change w by z in the truth table.

Page 8 – Change (Fig. 4e and g) by (Fig. 4d and g)

Page 10 – please check consistency between main text and different panels of Fig. 5.

Responses to Reviewers' comments:

Reviewer #1 (Remarks to the Author):

The manuscript "Implementing digital computing with DNA-based switching circuits" by Chunhai Fan and co-coworkers report the design of DNA switching circuits based on Strand displacement reactions. The field of DNA computing is exploding but, as the authors state in the introduction, there are still several problems to be faced. The authors try to overcome some of them by re-adapting the idea of switching circuits widely used in telecommunication. The idea is thus very interesting and quite novel. The experiments performed are very well described and convincing. Considering the topic and novelty I support the publication of this manuscript. Before that, I would suggest the authors to take care of the following minor issues:

1) Authors should make an effort to better clarify how their approach is improving the speed and modularity of DNA circuits. There is a nice section in the conclusion about this but my opinion is that this could be improved for clarity.

Response:

We thank the reviewer for this constructive suggestion. To better clarify the advantages of our DNA-based switching circuits (DSCs) in improving the speed and modularity, we have reorganized the discussion part and included new results in this revised manuscript.

1. We have added a discussion about the modularity of this approach, which is:

“The modularity can be reflected in molecular structure design, sequence design and DSC layout generation. The rule for molecular structure design is simple: an upstream switch only has a S domain while a downstream switch has a C domain and a S domain. Sequence design constraints of C domain are determined by the adjacent connection patterns (Supplementary Fig. 8). The DSC layout is generated by modularly mapping input combinations in the truth table to a transmission pathway on switch canvas.” (page 6, line 12-17)

We have also added a schematic illustration showing the constraints for modular design of strand sequences in Supplementary Figure 8.

Supplementary Figure 8. Modular sequence design for possible adjacent connection patterns within a DSC. a, For the pattern that an upstream switch (u) followed by one downstream switch (d), the sequence design constraint is determined by $C(d)=cs(u)$, where $C(d)$ means C domain of d and $cs(u)$ means current signal from u. b, For a fan-in pattern that n upstream switches are followed by one downstream switch, the constraints are determined by $C(d)=cs(u1)=cs(u2)=\dots=cs(un)$. c, For a fan-out pattern that one upstream switch is followed by n downstream switches, the constraints are determined by $C(d1)=C(d2)=\dots=C(dn)=cs(u)$. d, For a pattern that n upstream switches are followed by m downstream switches, the constraints are determined by $C(d1)=C(d2)=\dots=C(dm)=cs(u1)=cs(u2)=\dots=cs(un)$.

2. We have rewritten the section regarding the speed improvement with our DNA switching circuits. The

speed is improved from the following aspects: short single switch flipping time, high-speed inter-layer signal transmission, minimal circuit layers and reduced unwanted binding with less DNA strands (page 6, line 27-37).

In addition, we have provided a new figure in the SI showing the structural optimization of a downstream switch molecule to achieve fast current signal transmission and high specific switch flipping with low leakage.

Supplementary Figure 6. Structural optimization of the downstream switch. A downstream switch contains a C domain for current signal and a S domain for switching signal, which is formed by hybridization of three strands. The required features are quick replacement of S3 by current signal and low leakage from replacement of S2 by switching signal at the absence of current signal. We found a 0 nt gap led to slow reaction and a 2 nt gap led to higher leakage. We chose 1 nt as the gap length which has low leakage similar to the 0 nt gap and high reaction rate identical to the 2nt gap.

2) Authors should provide more experimental details in the main text. Now it seems like the concentration of the switches and inputs is not important as this is not stated during the discussion of the results. This should be more carefully described and discussed as represents an important factor in the response observed.

Response:

We thank the reviewer for pointing out this important issue. In the revised manuscript, we have added descriptions about the experimental details in the main text (page 3, line 33-38 and page 3, line 42 to page 4, line 4). To better explain the choices we have made about concentration, we have provided new results showing the influence of concentration ratio from different aspects (the ratio between strands within a single switch, the ratio between input and switch and the ratio between an upstream switch and a downstream switch) in the Supplementary Figures (Supplementary Figure 1, 2, and 3).

Supplementary Figure 1. Optimization of ratio between strands within a switch. a, Switching performance with different concentration ratios between S1 and S2 in a starting switch. A higher concentration of S1 ensures complete hybridization of S2, decreasing signal leakage. As 1.1X is sufficient for suppressing leakage, we used S1:S2=1.1:1 for experimental tests. **b,** Switching performance with different S3 concentrations in a downstream switch. Excessive S3 helps suppress leakage caused by unwanted switch flipping by switching signal at the absence of current signal (red line). So we used the ratio S1:S2:S3=1.1:1:1.5 for experimental tests.

Supplementary Figure 2. Output and leakage with different ratios of upstream switch and downstream switch. To ensure that the upstream switch could produce sufficient output for the downstream switch, we used a ratio of 1.5:1.

Supplementary Figure 3. Switching performance with 1X and 2X inputs. For single-switch and layered-switch circuits, 2X input could result in quicker switch flipping and higher plateau. So we used 2X switching signals as inputs.

As suggested by the reviewer, the concentration of the switches and inputs is an important factor in the observed response. To better clarify the concentration setup, we have included a Supplementary Figure showing the concentration for all tested circuits (Supplementary Figure 5).

Supplementary Figure 5. Concentrations for the tested circuits. Here 1X means 100 nM

3) Authors should more clearly define the threshold levels used during their experiment

Response:

We thank the reviewer for raising this concern. As the leakage is very low during the short computation time, we did not use any thresholding circuits between switches. For the binarization and evaluation of computing results, we used 0.4X and 0.6X as thresholds. Signals below 0.4X were treated as 0 and signals above 0.6X were treated as 1. If an output signal was between 0.4X and 0.6X, we considered it as a confusing result. Fortunately, the results from all circuits we tested all fell within the correct range. We have added the definition of the threshold we used for all experiments in the methods section in the new manuscript (page 7, line 34-39).

4) It would be also nice to provide a direct comparison with conventional DNA-based circuits in the main text rather than in the supplementary section.

Response:

We appreciate the reviewer's suggestion and have rearranged the manuscript accordingly as following:

1. We have moved the schematic illustration showing the dual-rail logic gate circuit for the full-adder from SI to Figure 5c. With a direct comparison with the conventional logic gate circuit, the simplicity of switching circuit is more impressive in circuit layout and number of involved computing elements.

Fig. 5 | Implementing a full-adder with a DSC. a-d, Logic gate diagram (a), truth table (b), dual-rail representation (c) and switching circuit diagram (d) of the full-adder circuit. e, Fluorescence readout of sum (left) and carry (right) with all possible combinations of inputs.

2. We have implemented a new application to solve the square root of a four-bit number with DSC. The new results are provided in Figure 6 in the revised manuscript. In this figure, we included a direct comparison of our DSC with previous implementations based on logic gates in number of computing elements and number of participated DNA strands, which further shows the improved molecular efficiency using DSC to solve the same digital problem.

Fig. 6 | A DSC for square root calculation. a, The DSC used to calculate the square root of a four-bit

number. **b**, A dual-rail logic circuit to perform the same square-rooting function. **c**, Number of computing elements using DSC in comparison with previous logic gate circuits with seesaw gates and single-stranded gates. **d**, Number of participated DNA strands using DSC in comparison with previous logic gates implementations. **e**, Experimental computing kinetics with four representative inputs.

Responses to Reviewer #2 (Remarks to the Author):

Wang and colleagues build several DNA-based switching circuits with multiple layers using the concept of toehold-mediated DNA strand displacement reactions, in which a single-stranded DNA molecule displaces another from a double-stranded complex by binding to an exposed short toehold domain. In the proposed switching circuits, a released strand from a starting switch is transmitted to a downstream switch, which then exposes a toehold and can therefore respond to its own switching signal strand. The idea of using DNA switching circuits for molecular digital computing (inspired by Shannon's work on electronic circuits) had been briefly proposed by Qian and Winfree (2011, *J Royal Society Interface* 8:1281-1297) as a way to build large-scale circuits. In the current manuscript, the authors designed and tested experimentally these circuits by including a reporter that responds to the signal flow and generates a fluorescent output. The study was well conducted and could demonstrate a faster computing speed and lower signal leakage in relation to logic gate circuits.

Response:

We are thankful for the reviewer's positive comments on our work. In this revised version we have added a citation of Qian and Winfree's pioneering work and a statement of the scientific questions we addressed in this work (page 2, line 25-27). Compared to the previous work proposed by Qian and Winfree, we focus on experimentally implementing switching circuits with good performance and exploring the advantages for solving digital tasks using DNA-based switching circuits (DSCs).

However, since a similar framework to scale up and systematically create DNA circuits was suggested by other group in 2011, I would expect to see now some practical application of this research to fully meet with the high standards of *Nature Communications*.

Response:

We thank the reviewer for this constructive suggestion. In the revised manuscript, we have demonstrated the implementation of a four-bit square-rooting algorithm with a DSC. We tested all 16 possible inputs and the results are shown in **Figure 6** and **Supplementary Figure 12**. In this application, the simplicity with switching circuit architecture is further demonstrated in terms of circuit layers, required computing elements and involved DNA strands. We also observed high computing speed with a half-completion time of less than 10 min for all inputs, which showed, to our knowledge, the fastest computing with diffusive components. We have included a description of these new results and discussion in this revised manuscript (page 5, line 31 to page 6, line 7).

Fig. 6 | A DSC for square root calculation. a, The DSC used to calculate the square root of a four-bit number. **b**, A dual-rail logic circuit to perform the same square-rooting function. **c**, Number of computing elements using DSC in comparison with previous logic gate circuits with seesaw gates and single-stranded gates. **d**, Number of participated DNA strands using DSC in comparison with previous logic gates implementations. **e**, Experimental computing kinetics with four representative inputs.

Supplementary Figure 12. Computing kinetics for all four-bit inputs with the square-root circuit.

The leakage of DNA-based switching circuits is still around 5%. Could the authors discuss what might be the sources of this leakage.

Response:

Although we have made efforts in molecular design and experimental conditions to ensure efficient output and suppress signal leakage, a small amount is still unavoidable. We hypothesize the observed leakage may come from these sources: imperfect duplex formation, slow leak reaction caused by binding of switching signal without current signal and thermal fluctuation induced exposing of bound regions.

As the switches were directly used after annealing without further purification, there a possibility that the current signal is not fully bound, which may be caused by quantification errors or insufficient reaction. To suppress this kind of leakage, we used a bit excessive switch signal binding strand to ensure sufficient binding of current signal. We have included new results showing the influence of strand ratios within a switch in Supplementary Figure 1 in the revised manuscript. With the optimized ratios, although the leakage cannot disappear completely, it was efficiently suppressed.

Supplementary Figure 1. Optimization of ratio between strands within a switch. **a**, Switching performance with different concentration ratios between S1 and S2 in a starting switch. A higher concentration of S1 ensures complete hybridization of S2, decreasing signal leakage. As 1.1X is sufficient for suppressing leakage, we used S1:S2=1.1:1 for experimental tests. **b**, Switching performance with different S3 concentrations in a downstream switch. Excessive S3 helps suppress leakage caused by unwanted switch flipping by switching signal at the absence of current signal (red line). So we used the ratio S1:S2:S3=1.1:1:1.5 for experimental tests.

As shown in the new included Supplementary Figure 6, the downstream switch we used had a gap, so the switching signal may turn on the switch without the current signal from an upstream. We have optimized the gap length to 1 nt which has fast reaction rate and very low leakage. However, there is still a possibility that the switching signal replaces the current signal (S2) very slowly via this 1 nt toehold.

Supplementary Figure 6. Structural optimization of the downstream switch. A downstream switch contains a C domain for current signal and a S domain for switching signal, which is formed by hybridization of three strands. The required features are quick replacement of S3 by current signal and low leakage from replacement of S2 by switching signal at the absence of current signal. We found a 0 nt gap led to slow reaction and a 2 nt gap led to higher leakage. We chose 1 nt as the gap length which has low leakage similar to the 0 nt gap and high reaction rate identical to the 2nt gap.

We have also provided new results showing the influence of temperature on the operation of switching circuits in Supplementary Figure 7. Changing temperature from 20 °C to 35 °C did not have significant influence on small circuits that have one or two switches, but slightly increased the leakage of a fan-in circuit when the downstream switching signal was added. This suggests a higher thermal motion could increase the binding of switching signal without the arrival of current signal. Despite this, we could see the leakage remained at a low level below 20%, providing a high thermal stability for potential implementation in vivo.

Supplementary Figure 7. The influence of reaction temperature on the operation of DSCs. Increasing temperature from 20 °C to 35 °C did not have significant influence on the leakage and output of the signal-switch and two-switch circuits. Despite the leakage for a fan-out circuit slightly increased with the temperature, it remained at low level.

Minor points

Fig 3, panel f – change w by z in the truth table.

Response:

We thank the reviewer for pointing out this mistake. We have corrected this in Figure 3f in the new version.

Page 8 – Change (Fig. 4e and g) by (Fig. 4d and g)

Response:

We have changed “Fig. 4e and g” to “Fig. 4d and g” in the main text (page 5, line 9).

Page 10 – please check consistency between main text and different panels of Fig. 5.

Response:

We have carefully checked the consistency between main text and the panels of Fig.5. In the revised

manuscript, we have corrected the inconsistency between the referred panels and the panels in Fig.5, which are highlighted in the main text (page5, line 19-25).

Reviewers' Comments:

Reviewer #1:

Remarks to the Author:

The authors have responded to all my comments. I am ok with publication of the manuscript in the present form.

Reviewer #2:

None